# Exploration of street space architectural color measurement based on street view big data and deep learning—A case study of Jiefang North Road Street in Tianjin

**Xin Han**[1☉]*, **Ying Yu**[2☉], **Lei Liu**[3], **Ming Li**[4], **Lei Wang**[5]*, **Tianlin Zhang**[5]*, **Fengliang Tang**[5]*, **Yingning Shen**[6]*, **Mingshuai Li**[5]*, **Shibao Yu**[5,7]*, **Hongxu Peng**[8]*, **Jiazhen Zhang**[5]*, **Fangzhou Wang**[9], **Xiaomeng Ji**[10], **Xinpeng Zhang**[11], **Min Hou**[12]

**1** Department of Landscape Architecture, Kyungpook National University, Daegu, South Korea,
**2** Department of Landscape Architecture, College of Forestry, Shandong Agricultural University, Taian, China, **3** School of Architecture, Harbin Institute of Technology, Shenzhen, China, **4** Gengdan Academy of Design, Gengdan Institute of Beijing University of Technology, Beijing, China, **5** School of Architecture, Tianjin University, Tianjin, China, **6** School of Cultural Heritage, Northwest University, Xi'an, China, **7** School of Architecture and Urban Planning, Lanzhou Jiaotong University, Lanzhou, China, **8** School of Architecture and Urban-Rural Planning, Fuzhou University, Fuzhou, China, **9** Chengdu Tianfu New Area Institute of Planning & Design Co., Ltd, Chengdu, China, **10** Department of Tourism, Management College, Ocean University of China, Qingdao, China, **11** Landscape Architecture Research Center, Shandong Jianzhu University, Jinan, China, **12** Fuzhou Planning & Design Research Institute Group Co. Ltd, Fuzhou, China

☉ These authors contributed equally to this work.
* wanglei2021@tju.edu.cn (LW); zhangtianlineric@tju.edu.cn (TZ); tang_fengliang@tju.edu.cn (FT); 202210097@stumail.nwu.edu.cn (YS); limingshuai.lee@gmail.com (ML); yushibao@tju.edu.cn (SY); penghongxu@fzu.edu.cn (HP); zhangjz_2015@tju.edu.cn (JZ)

**Data Availability Statement:** Data are available at https://doi.org/10.6084/m9.figshare.22250176.v1.

## Abstract

Urban space architectural color is the first feature to be perceived in a complex vision beyond shape, texture and material, and plays an important role in the expression of urban territory, humanity and style. However, because of the difficulty of color measurement, the study of architectural color in street space has been difficult to achieve large-scale and fine development. The measurement of architectural color in urban space has received attention from many disciplines. With the development and promotion of information technology, the maturity of street view big data and deep learning technology has provided ideas for the research of street architectural color measurement. Based on this background, this study explores a highly efficient and large-scale method for determining architectural colors in urban space based on deep learning technology and street view big data, with street space architectural colors as the research object. We conducted empirical research in Jiefang North Road, Tianjin. We introduced the SegNet deep learning algorithm to semantically segment the street view images, extract the architectural elements and optimize the edges of the architecture. Based on K-Means clustering model, we identified the colors of the architectural elements in the street view. The accuracy of the building color measurement results was cross-sectionally verified by means of a questionnaire survey. The validation results show that the method is feasible for the study of architectural colors in street space. Finally, the overall coordination, sequence continuity, and primary and secondary hierarchy of

**Funding:** Young Scholars Science Foundation of Lanzhou Jiaotong University (2020033). In this study, the sponsor Hongxu Peng participated in the study design and data analysis. The funding is Fujian Provincial Social Science Planning Project "Study on the Influence and Significance of Zhu Zi Studies on Korean Confucian Habitat Culture"(Grant No.FJ2021C035).

**Competing interests:** The authors have declared that no competing interests exist.

architectural colors of Jiefang North Road in Tianjin were analyzed. The results show that the measurement model can realize the intuitive expression of architectural color information, and also can assist designers in the analysis of architectural color in street space with the guidance of color characteristics. The method helps managers, planners and even the general public to summarize the characteristics of color and dig out problems, and is of great significance in the assessment and transformation of the color quality of the street space environment.

## Introduction

Architectural color is an intuitive reflection of the city's natural characteristics and regional culture, and a true expression of the quality of the urban environment [1]. The determination of the current state of architectural color is both a respect for the natural characteristics of the city and regional culture, and a basic way to analyze the current problem, which is directly related to the renewal and renovation of architecture as well as new construction [2]. Color in the interior of buildings has also been studied by some scholars [3]. Although the urban space architectural color measurement has achieved some success in recent years, but limited by the longtime of architectural color data collection, excessive human and material resources consumption and other limitations, the mainstream color measurement method is still mainly quantitative research, but also part of the quantitative research using a combination of local architectural color sampling and expert interviews [4]. Large-scale research and quantitative analysis can reveal the current characteristics of urban architectural color more objectively, weaken the interference of subjective consciousness in qualitative research, and then dig out the characteristics and shortcomings of urban architectural color, and put forward more targeted solutions and suggestions [5]. Therefore, the measurement of architectural color on a large coverage and with high efficiency is a fundamental and important part of urban color research [6]. The exploration of effective large-scale data acquisition and quantitative analysis technology has attracted extensive attention in the academic circle.

With the convenient data access and the development of information processing technology, APIs provided by map services for scholars and researchers to extract spatially high-resolution images of streets and neighborhoods are increasingly used in interdisciplinary research, providing scientific support for the development of different disciplines [7–9]. Scholars and research institutions have done a lot of exploration on quantitative urban research based on street view images [10, 11]. Zhang used Google Street View in combination with various open source data to identify mechanisms that affect the built environment of cities [12], Han used Google Street View images combined with machine learning techniques to make predictions about the perception of stress in city streets [13], Wang explored different perception conditions of urban streets using Baidu Street View images combined with spatial syntax [14], Yao explores six different perceptions of urban street conditions by combining street view images with human-machine adversarial models [7]. Using big data and deep learning technology of street view images has overcome the difficulties of using traditional data for street space research to a certain extent, and has led to changes in the perspective, conditions, scales, and methods for street research [15–19]. Despite the fact that many scholars have conducted some research and exploration on urban space using street view images, the use of street view image big data combined with deep learning to measure the color of urban space architectural research is still in the initial stage.

Baidu Maps was found to have better coverage of the Chinese region than other maps, allowing for a more detailed study of the Chinese study area [13]. In this study, Baidu Street View was used as the data source, and Jiefang North Road in Tianjin was chosen as the research space for urban architectural color measurement. The study was conducted from the selection of the research object to the selection of the color extraction method, and finally to the color expression and its application. The research aims of this study were twofold. (1) The construction of a whole process of street space architectural color measurement based on street view big data and deep learning technology was introduced, encompassing data set and neural network selection, neural network construction and training, and color extraction and analysis. This was done to provide technical support for the objective quantitative analysis of street space architectural color by validating and reflecting on the results. (2) The analysis of the color of the street space of Jiefang North Road in Tianjin was conducted, summarizing the advantages and potential problems of the current color. Targeted design guidelines and optimization suggestions were proposed based on the results of color attribute analysis. The joint analysis method of Street View Big Data and deep learning was implemented, providing researchers and urban planners with more targeted data on architectural color perception in urban space, and advancing urban planning practice.

## Methodology

**Fig 1** shows the conceptual framework of architectural color measurement in urban space from a holistic perspective. Firstly, we download the road network data of Tianjin North Road according to the study area, take a street point every 20 meters on the road network, adjust the street view image acquisition parameters by simulating the pedestrian's perspective, apply the Baidu Map API to collect the urban street view data, set the street view saving location, and finally obtain all the street view images in the study area. The Cityscapes dataset was used to train the SegNet deep learning neural network, and then semantic segmentation was performed on the Street View images of the study area to obtain the visual element data of the city streets. In order to separate the building elements from the image, the non-building visual elements of the semantic segmented street view image are masked, and then the edges of the building elements are optimized to highlight the edges of the building elements. Volunteers were invited to evaluate the feasibility of architectural colors identified by K-Means algorithm to prove the effectiveness of the method. Finally, a summary analysis of the fundamental colors of Tianjin North Road was conducted, the saturation of the fundamental colors and their color value in the city streets were analyzed, and the overall coordination, sequence continuity and

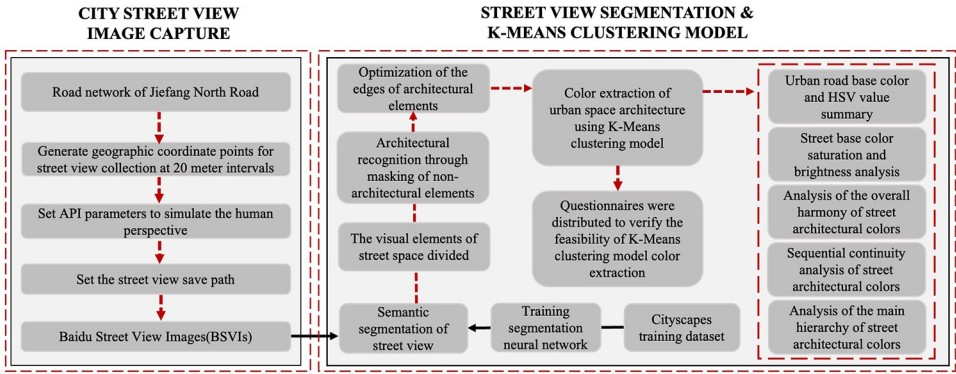

**Fig 1. Overview of workflow.**

main hierarchy of the street architectural colors were analyzed. Based on the results of the analysis, we propose design guidelines and optimization suggestions.

## Study area

Jiefang North Road Street is located in the eastern part of the Heping District of Tianjin, near the west bank of the Haihe River, from Jiefang Bridge in the northwest to Xuzhou Road in the southeast and is the most historical financial street in Tianjin. Since the opening of Tianjin to commerce in 1860, the Jiefang North Road Street was the first to be developed and has experienced the peaks and valleys of prosperity-depression-boom-depression. In 2019, the Tianjin Municipal Planning Bureau proposed a new plan for the Jiefang North Road Street area, proposing to revitalize the stock of architecture, update the commercial ecology, improve the quality of the street space, enhance urban vitality, and develop it into a high-end service industry gathering area featuring tourism and leisure, business and finance, on the basis of historical culture, transportation hub resources and historical architectural preservation. In addition, there are 9 roads in Jiefang North Road Street, including Jiefang North Road, which is known as the "Oriental Wall Street". Jiefang North Road is one of the priority style control areas in Tianjin, but in the process of repeated construction and renovation, it is strictly required to protect the integrity and authenticity of the street architecture and ancillary architectural features, so that the style is intact. The classical style of stately and stable appeal and the imitation of architectural materials such as reinforced concrete, marble, granite and brick red ensure the stability of the overall color of the street. Therefore, the Jiefang North Road, which has a high reputation, was selected for the specific operation of architectural color measurement and validation of the color measurement results.

## Street view data acquisition and semantic segmentation

**BSVIs data collection.** In this study, we used the structure of "one vertical and eight horizontal" in the latest planning to study the architectural color of the street space in Jiefang North Road area, where one vertical is Jiefang North Road and eight horizontals are Changchun Road, Binjiang Road, Harbin Road, Chifeng Road, Chengde Road, Yingkou Road, Datong Road and Dalian Road (**Fig 2**). Street View data allows perception and observation of the urban environment from a human-centered perspective [20]. The Street View data platform not only provides street view browsing services for web users, but also publishes an application program interface (API). According to the panoramic static map document (https://lbsyun.baidu.com/index.php?title=viewstatic) in the web service API in Baidu Maps Open Platform, we can Submitting the appropriate parameters to call the API for Baidu Street View map acquisition. After reviewing the site and reviewing Baidu Maps' data policy, we confirm that we are in compliance with Baidu's terms of service [21]. In order to obtain the street view images from the pedestrian perspective along the road in the study area of Jiefang North Road in Tianjin, the parameters for the street view big data acquisition need to be set [22]. In order to ensure the low repeatability and perceptibility of the sampled data, a sampling distance of 20m was chosen to obtain the street view data collection points, and the same angle of street view image data was acquired in the sequence of human moving. The starting and ending coordinates of street view images collected from different roads are shown in **Table 1**. The URL example for collecting street views is as follows: http://api.map.baidu.com/panorama/v2ak=YOUR_KEY&width=1024&height=512&location=120.219699226437,30.203814207756&pitch=20&fov=150&heading=90.

In this URL, 'ak' is the key for the map plane, 'width' is the image width, ranging from 10 to 1024 pixels, and 'height' is the image height, ranging from 10 to 512 pixels. 'Fov' is the

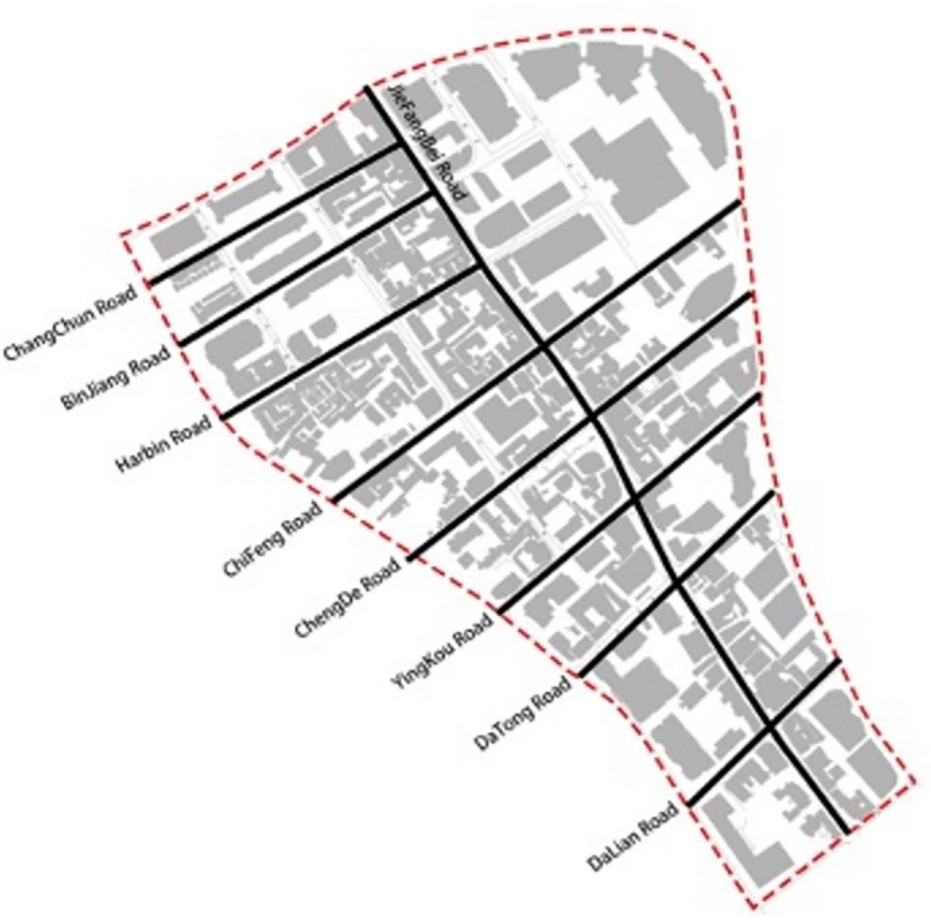

**Fig 2. Jiefang North Road Street.**

horizontal distance, and the horizontal viewing angle is set to 150 degrees, based on human visual habits. 'Heading' is the horizontal angle, set to 90 degrees after multiple experiments. 'Location' is the spatial geographic coordinates of the image data to be submitted to the server.

**Deep learning-based image segmentation.** In order to better identify the color of architectural street space, this study uses SegNet network structure based on ResNet backbone

**Table 1. The starting and ending points and sampling points of each street scene.**

| Road Name | The starting of road | The ending of road | Image sampling points |
|---|---|---|---|
| Jiefang North Road | 117.213152,39.138209 | 117.220397,39.12924 | 52 |
| Changchun Road | 117.209886,39.135731 | 117.213795,39.13756 | 18 |
| Binjiang Road | 117.210378,39.135022 | 117.215779,39.137731 | 29 |
| Haerbin Road | 117.211195,39.134227 | 117.214785,39.136038 | 21 |
| Chifeng Road | 117.212587,39.133096 | 117.217291,39.13602 | 26 |
| Chengde Road | 117.213788,39.132452 | 117.218783,39.135667 | 29 |
| Yingkou Road | 117.219071,39.134629 | 117.21506,39.13174 | 24 |
| Datong Road | 117.216219,39.131056 | 117.219246,39.133449 | 19 |
| Dalian Road | 117. 220214,39. 13136 | 117.218004,39.129579 | 14 |

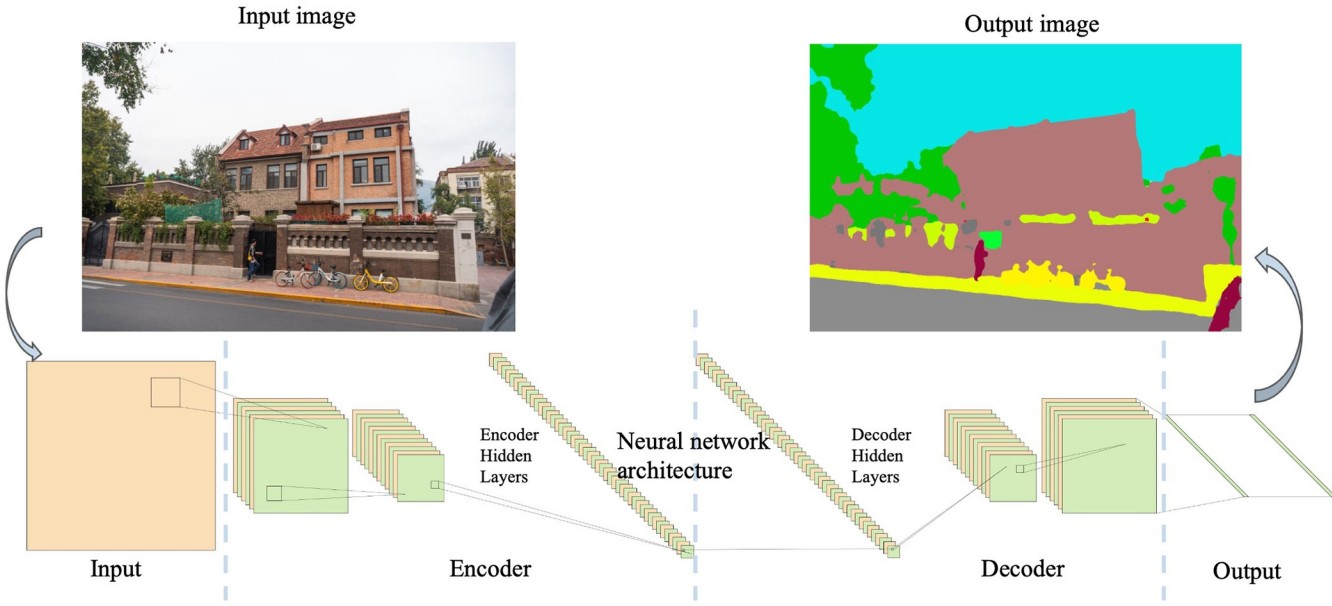

**Fig 3. SegNet architecture.**

model for training, validation and prediction of the dataset. SegNet (**Fig 3**) is an open-source project for image segmentation developed by a team at the University of Cambridge. It is a type of convolutional neural network with the main motivation of efficient scene understanding and can segment images at the pixel level based on the information of object features in the image, and is widely used in various industries for scene understanding applications [14]. The core structure consists of an encoder network, a corresponding decoder network and a classification layer. The encoder follows the network model of VGG16, which mainly parses the objects in the image scene according to their labels, and the decoder corresponds the parsed information to the final image form, with each pixel using a color corresponding to its object information [23–25].

By analyzing different datasets, the CityScapes data were finally selected for neural network training. CityScapes data was originally used for car self-driving training, and the scenarios include 50 different cities, daytime scenes in spring, summer and autumn, and a variety of scene such as video and pictures in addition to rain and snow [26]. The dataset contains 5000 pixel-level finely labeled images of street scenes, which are divided into three parts: training, validation, and test sets. The training and validation sets are used to jointly adjust the model parameters to arrive at the optimal model, and the test set is used to measure the generalization ability of the optimal model in practice. Of these, 2975 are used for training, 500 for validation, and 1525 for testing. The partitioning criteria ensure that each partitioned dataset contains various scenarios to ensure that the training results can be used to predict a variety of scenarios, and the labels are divided into 30 categories, which are mostly used by professionals in street space research due to their large data quantities, rich scenarios, fine labeling and easy availability of open source [13, 14, 22]. In this study, the 5000 finely labeled images are simplified into two parts, training and validation, and the labels are reclassified into 19 categories according to the actual needs. The training results show that the accuracy in the training set is 90.83% and the accuracy in the validation set is 89.95%. This accuracy can achieve the work of image semantic segmentation effectively.

### Street view data cleaning—architecture in the spotlight

**Masking of non-architectural elements for architectural recognition.** After semantic segmentation of the sample street view image, in order to separate the building elements from the image content to avoid the interference of a large number of vegetation, vehicles and other elements on the architectural elements, the remaining part of the street view image needs to be masked. The image mask is to control the area of image processing by masking the street view image to be processed with selected images or objects. Its main purpose is to extract the architectural area in the street view image, and multiply the pre-made mask of the architectural part (the prediction result) with the image to be processed (the original image) to get the image of the architectural part, so that the image value of the architectural area remains unchanged, while the image values outside the area are all 0 (solid black) (**Fig 4**), so as to achieve the purpose of architectural recognition.

**Optimization of the edges of architectural elements.** After the mask processing, it can be found that the semantic segmentation result has enough accuracy for architectural recognition, but there are more fragmented parts on the edges. In order to achieve more detailed architectural recognition, it is necessary to process the fragmented parts on the edges of the architectural edges, and it is found through the study that the erosion and expansion algorithms are useful for architectural edge processing. In order to further improve the recognition accuracy of architectural edges, the closed operation (expansion followed by erosion) algorithm in OpenCV is adopted to process architectural edges and achieve the optimal extraction of architectural elements.

### Extraction and expression of architectural colors

The growing use of image research in urban color surveys has emphasized the significance of extracting image color features, particularly dominant colors, which greatly impact visual perception. The K-Means clustering algorithm is an especially suitable method for extracting dominant colors in architectural images, as it efficiently groups similar color data points in the color space [27]. As an unsupervised learning technique, the algorithm's objective is to divide a set of observations into K clusters, with each observation belonging to the cluster with the nearest mean or centroid. The K-Means clustering process involves several key steps to achieve convergence. First, initial centroids are selected, typically by choosing K random data points from the dataset. Next, data points are assigned to the nearest centroid, forming K clusters

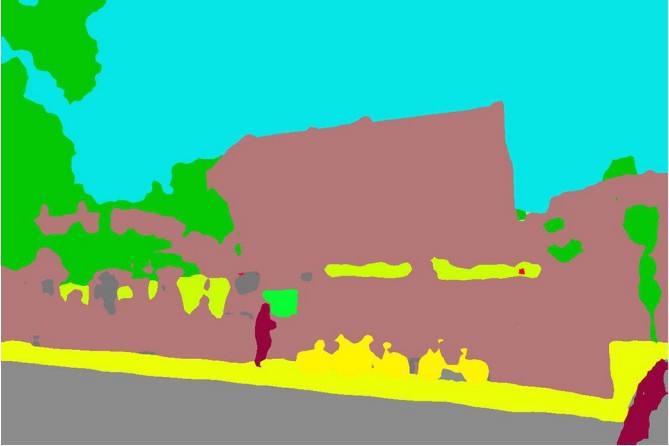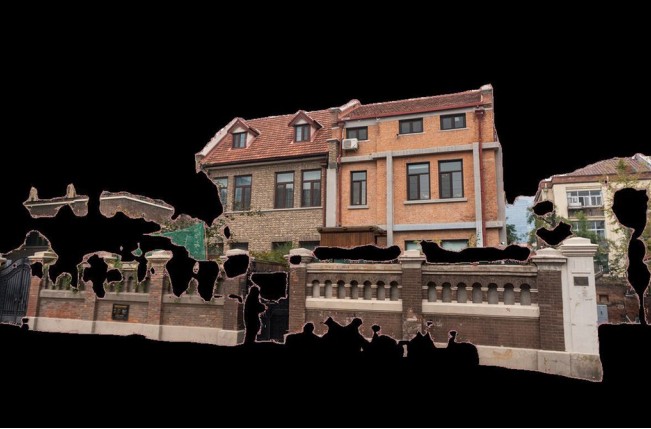

**Fig 4. Non-building element mask processing result schematic.**

based on their proximity to these centroids. Once clusters are formed, the centroids are updated by calculating the mean of all data points within each cluster, resulting in new centroid positions. This iterative process of reassigning data points to the nearest centroid and updating centroids continues until a specified convergence criterion is met, such as when the centroids' movement falls below a certain threshold or a maximum number of iterations is reached. Upon convergence, the final centroid coordinates of each cluster represent the dominant colors in the architectural image.

Architectural color research primarily examines street view images featuring architectural elements, which often display color distribution in block-filled patterns and have a relatively concentrated pixel distribution. When applying the K-Means clustering algorithm to extract dominant colors, the position relationship of pixels in the image is not taken into account, and the primary clusters run in the color space. This color set extraction method, based on the K-Means clustering algorithm, is widely employed in urban color research, as it enables the customization of the desired number of color categories (K) and iterations, ultimately identifying the dominant colors present in the architectural images. Therefore, the method of color set extraction based on the K-Means clustering algorithm is widely adopted in research within the field of urban color.

## Results

### Feasibility verification of architectural color extraction based on K-Means clustering algorithm

In the process of cluster color extraction using K-Means, after several simulations, it was found that when the number of clusters was 8, it was closer to the color abstraction of human visual space (**Fig 5**), therefore the number of clusters was specified as 8 in this study. In order to ensure the objectivity and accuracy of the extracted architectural colors, this study performed color clustering on the images twice, firstly for a single image and secondly for multiple images. When extracting the color of a single image, the 7 colors corresponding to each image (except black) obtained by clustering can best represent the characteristic color of that sampling point. The color with the largest proportion can represent the dominant color of the viewpoint, and the color with the most prominent hue can be initially considered as the embellishment color of the viewpoint. The embellishment color is also the most easily found color in

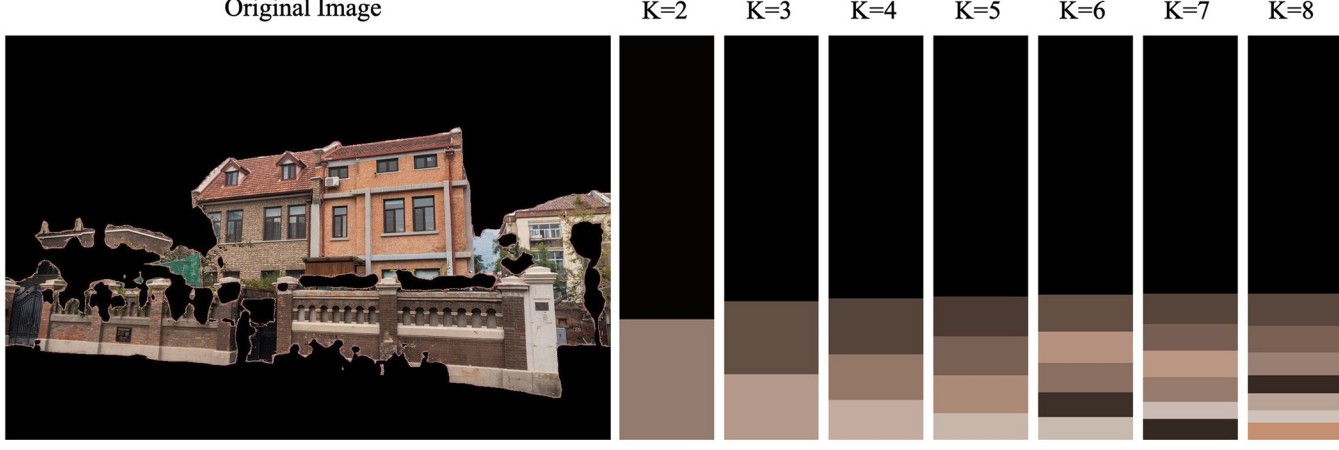

**Fig 5. Colors with different number of clusters.**

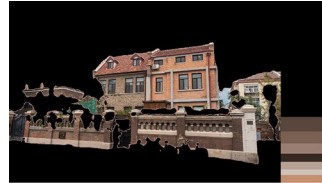
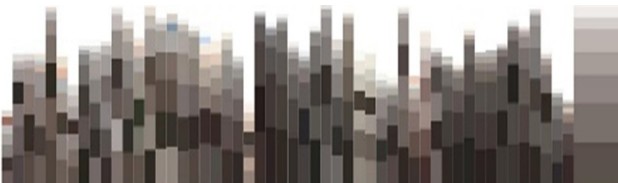

| Color Clustering of Single Street View | Basic color matching extraction results of multiple street view images bases on Jiefang North Road |

**Fig 6. Extraction and expression of architectural color.**

the whole street. The color extraction results basically match the color perception of the original image. The color extraction of multiple images is based on a large number of samples of a certain class of images to extract the color elements that are representative of the overall visual intention of the class of images. The principle is the same as that of the single image extraction method, and the final extracted color is obtained by the second K-Means clustering of the clustering results of the single image, and the color extracted by the second clustering can better express the comprehensive characteristics of the whole street (Fig 6).

In order to better investigate the consistency between the color extraction results and the human eye recognition results of the sampled images, five of them were randomly selected for a color comparison questionnaire survey. 120 online questionnaires were distributed and all of them were effectively collected. The demographic data of the survey respondents are shown in Table 2. The average age of the volunteers was 35.30 years, with a higher proportion of males (53.73%). In terms of their educational background, 27.47% completed primary school or below, 31.87% college or above, and 40.66% high school. Han Chinese was the most common ethnic group (92.12%), and local residents accounted for 83.73% of the volunteers. According to the survey results, the average proportion of color extraction results that are "completely consistent" with the original image is 46.5%, the average proportion that are "basically consistent" is 42%, the average proportion that are "basically inconsistent" is 10.83%, and the average proportion that are "completely inconsistent" is 0.66% (Fig 7). This result basically justifies the use of color model extraction results.

## Analysis of the dominant color of roads on Jiefang North Road Street

The results of the dominant color characteristics of streets in Jiefang North District show that the architectural colors of the streets in Jiefang North Road have a strong harmonious unity, which is the characteristic of Jiefang North Road as a historical style district, among which

**Table 2. Descriptive statistics for the volunteers.**

| Variables | | Proportion/Mean (SD) |
|---|---|---|
| Gender (%) | Male | 53.73 |
| | Female | 46.27 |
| Average age | | 35.30 (13.21) |
| Education (%) | Primary school or below | 27.47 |
| | College and above | 31.87 |
| | High school | 40.66 |
| Race (%) | Han Chinese | 92.12 |
| | Others | 7.88 |
| Residence (%) | Local resident | 83.73 |
| | Non-local resident | 16.27 |

| No. | Color extraction results | Statistical results | | | |
|---|---|---|---|---|---|
| | | a | b | c | d |
| 1 | | 53.33% | 37.5% | 8.33% | 0.83% |
| 2 | | 41.67% | 48.33% | 9.17% | 0.83% |
| 3 | | 51.67% | 40.83% | 6.67% | 0.83% |
| 4 | | 35% | 42.5% | 21.67% | 0.83% |
| 5 | | 50.83% | 40.83% | 8.33% | 0% |
| | Average value | 46.5% | 42% | 10.83% | 0.66% |

**Fig 7. Extraction results and verification analysis of architectural color.**

Yingkou Road, Datong Road and Dalian Road have less than ideal tone color characteristics, and the overall coordination is not good but does not affect the expression of high color value warm gray characteristics in the whole district (**Fig 8**). In future works, priority can be given to the improvement of the overall harmony of Yingkou Road, Datong Road and Dalian Road.

## Analysis of architectural color characteristics of Jiefang North Road Street

**Analysis of saturation and value of dominant colors in urban streets.** From **Figs 9** and **10**, we can observe the continuity of the dominant color at the overall level of the Jiefang

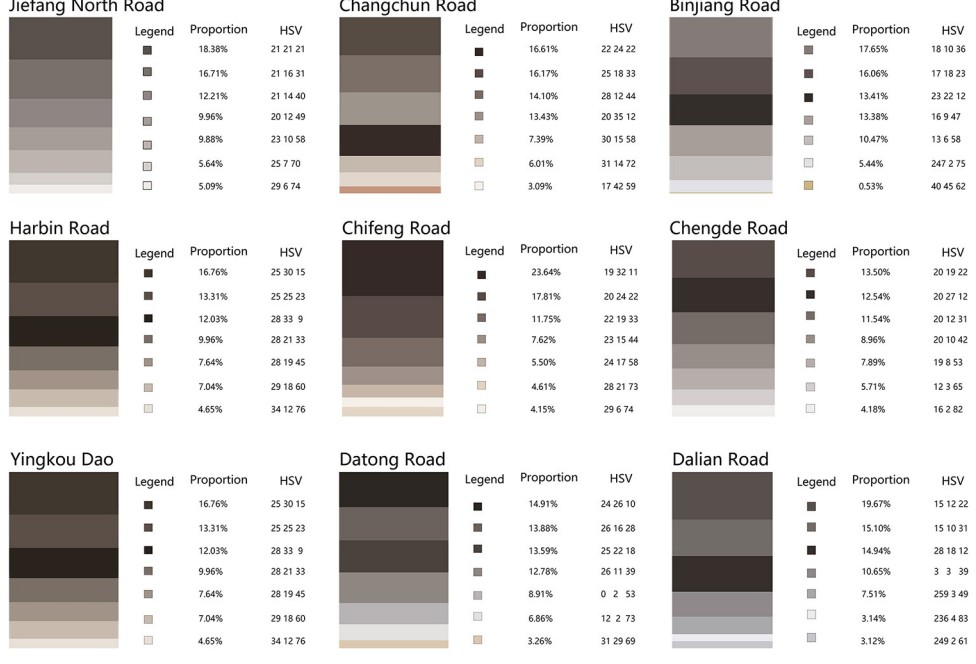

**Fig 8. Basic color of each road in the Jiefang North Road area.**

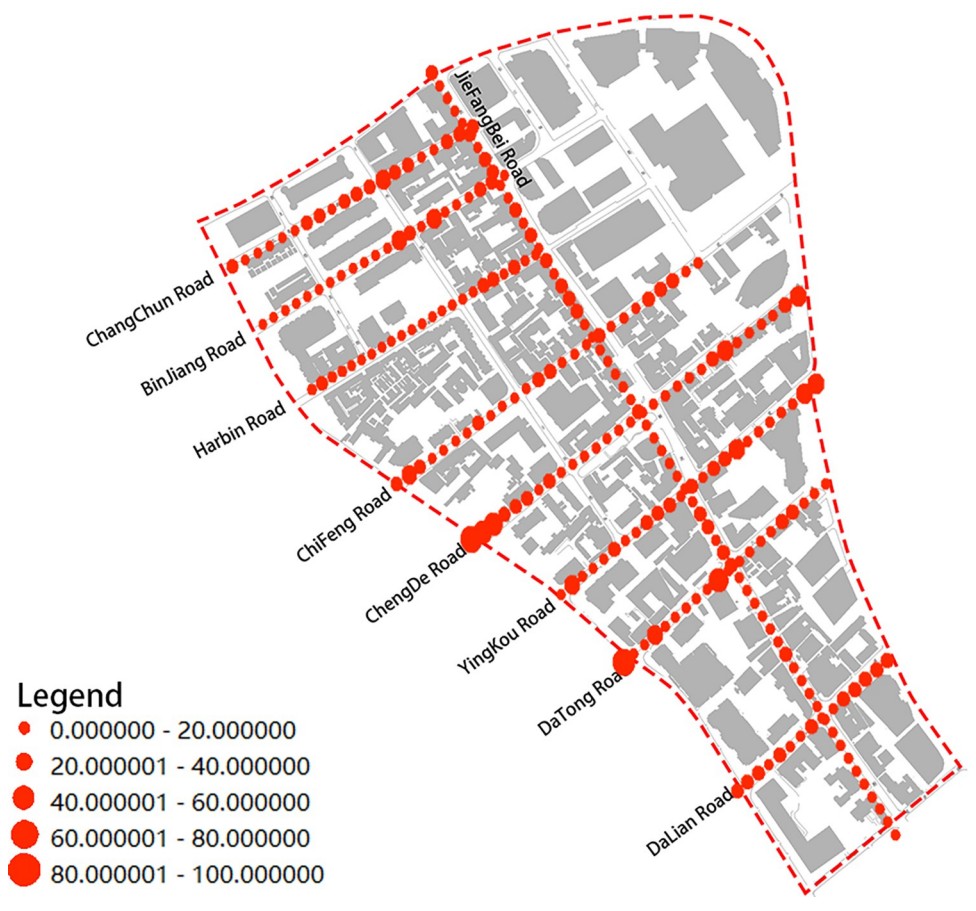

**Fig 9. Basic color saturation distribution map of the street.**

North Road Street. In **Fig 9**, it can be seen that the saturation of the dominant color fluctuates less, and there is some variation at road intersections. Road intersections have a superior location and are designed as important nodes in urban space. In order to enhance the recognizability of the architectural structure, developers often design the colors to enhance the node and landmark of the architectural structure. Therefore, it can be assumed that the saturation of the Jiefang North Road Street has a good serial continuity. Compared with the saturation distribution, the distribution of value cannot find obvious regular characteristics, and there is even a case that the contrast of value between two observation points reaches a strong contrast effect, so the sequence of color value of the architecture in the Jiefang North Road Street is poorly continuous (**Fig 10**). What exactly causes the change of color value needs to be further studied by integrating various factors affecting the architectural color expression.

**Analysis of the overall harmony of street architectural colors.** The H-S and H-V scatter diagrams of the dominant colors of the viewpoints were obtained by importing the colors into SPSS (**Fig 11**). The overall hue of the Jiefang North Road street is controlled between (0–60), saturation is controlled between (0–60), and value is controlled between (0–40), forming a low-saturation warm gray tone atmosphere in the whole road. It can be seen that architectural color design of Jiefang North Road is mainly based on similar color palette, with the overall use of similar color palette, and the color difference is controlled by chromaticity. The overall coordination of color is well controlled, and the overall style is simple, stable and unified, which is a good implementation of the architectural color control requirements of Tianjin's

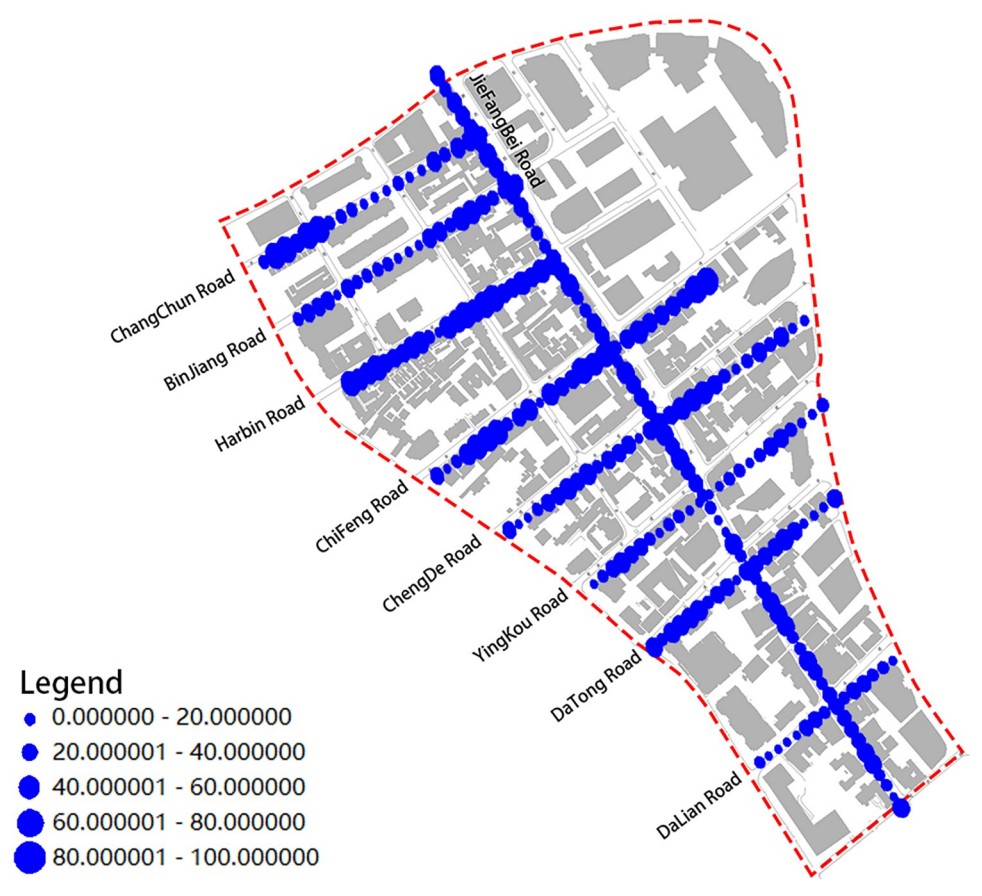

**Fig 10. Basic color value distribution map of the street.**

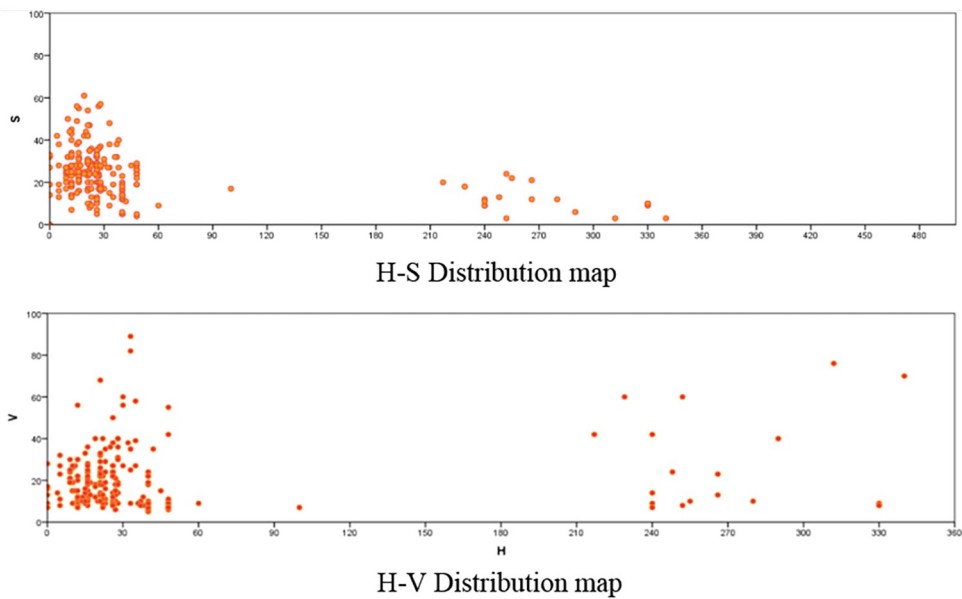

**Fig 11. Analysis of the overall coordination of street architectural color.**

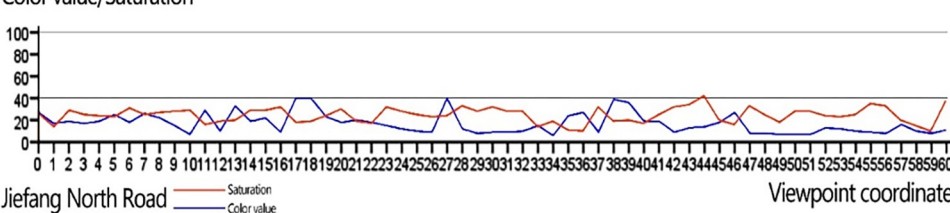

**Fig 12. Street sequence continuity analysis diagram of Jiefang North Road.**

historical conservation plan. Under the control of color, the material colors of red brick, green brick and marble can be well coordinated. Combining the H-V distribution and the map building color images, we can find that very few colors are prominent in the overall tone atmosphere, mainly affected by the value, although affected by the time of data collection, but also reveals the substantial problem of road intersections affecting the building color expression.

**Sequential continuity analysis of street building colors.** In the past, the analysis of color continuity was carried out in two ways: subjective judgment and quantitative analysis. The subjective judgment can be achieved by abstract street color sampling chart, and the quantitative analysis can be achieved by drawing a table of color continuity changes in color value and saturation. The series of operations of the metric model can realize the input of streetscape image data to the output of color elements, which provides support for the quantitative analysis of sequence continuity. After inputting the color data elements and viewpoint coordinates into the SPSS analysis software for line plotting, the saturation and color value indexes of the viewpoint sequence were connected to the points, and the changes of the main color of each street were summarized through the ups and downs of the line.

*(1) "One vertical" road sequence continuity analysis.* From the line graph of the continuity of the main color saturation and luminance of the Jiefang North Road sequence, we can see that the main color saturation and color value of Jiefang North Road are maintained between (0–40), and the continuity of the sequence is good. In the color value continuity diagram, the contrast between the color value of No. 27 and the surrounding area is large. (**Fig 12**).

*(2) "Eight horizonta" road sequence continuity analysis.* We can visually see the changes of architectural colors in the "eight horizontal" roads, and we can arrange the sequence continuity of the eight roads and find that the sequence continuity of Harbin Road, Chifeng Road, Dalian Road and Yingkou Road is better, while the color continuity of Changchun Road, Binjiang Road, Chengde Road and Datong Road is worse. (**Fig 13**) Specifically, the saturation and luminosity of the main colors of the Harbin buildings are controlled between 0 and 40, with no large sudden changes and good continuity. Like Harbin Road, the saturation and color value of Chifeng Road are controlled in the small-scale, continuous spatial scale of the street, and although the saturation fluctuates to a certain extent, the rhythm of change is stable and consistent, with less impact on the perception of visual continuity; Dalian Road has a high continuity of color value, although the color level gradually decreases due to the presence of the former Macquarie Bank building, but the road length is short, and the Macquarie Bank building at the intersection of the road is larger than the surrounding buildings, and is an important historical building, so it plays a leading role in the color of the street, so its visual continuity is not greatly affected, Yingkou Road has a small change in color level, but color value due to the high reflectivity of the building materials.

Compared with Harbin Road, Chifeng Road, Dalian Road and Yingkou Road, the sequence continuity of Changchun Road, Binjiang Road, Chengde Road and Datong Road is relatively

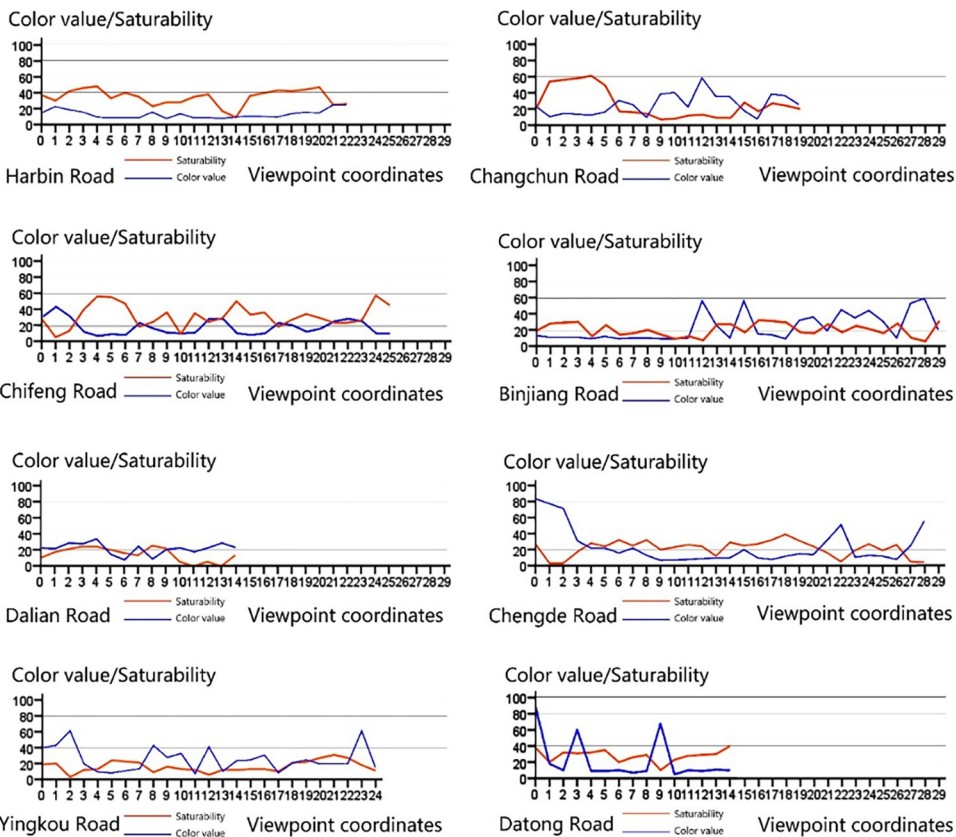

**Fig 13. Street sequence continuity analysis diagram of Jiefang North Road.**

poor. The color difference between the viewpoints before and after viewpoint 6 is close to 5. The reason for this is that the construction materials on both sides of the road are red bricks, red bricks combined with cement mortar finish bricks, and cement mortar finish bricks, with the boundaries being clearly defined. the main reason for continuity. The saturation continuity of Riverside Road is good, while the color value continuity is poor, and two abrupt changes are formed at viewpoints 11 and 15. The reason is that before viewpoint 11, there were high-rise buildings such as Huijin Center and Deyou Real Estate on the south side, which blocked a large area of the street space, while after viewpoint 11, the building height was reduced, and the material of the exterior wall of Rujia Hotel chose bright yellow finish tiles and large area of striped glass, which caused a sudden change of color value due to the high reflectivity of the material. The alternate appearance of a certain pattern, its color value and saturation of change tends to stabilize. Datong Road has good color continuity and poor color value continuity, with two sudden changes in color value due to the incompleteness of the building interface and the presence of the former China-Russia Daosheng Bank building. Chengde Road has good continuity of color value and saturability, but the intersection with Dagu North Road and Zhangzizhong Road is open, which causes sudden changes in color value.

On the whole, after analyzing the sequence continuity in terms of color value and saturability for more than 200 viewpoints in 9 roads, we can grasp the degree of color unity in different streets and visually determine the discordant elements that affect the sequence continuity. According to the results of the analysis, the visual continuity of Jiefang North Road is the best, followed by Harbin Road, Chifeng Road, Yingkou Road and Dalian Road, while Changchun

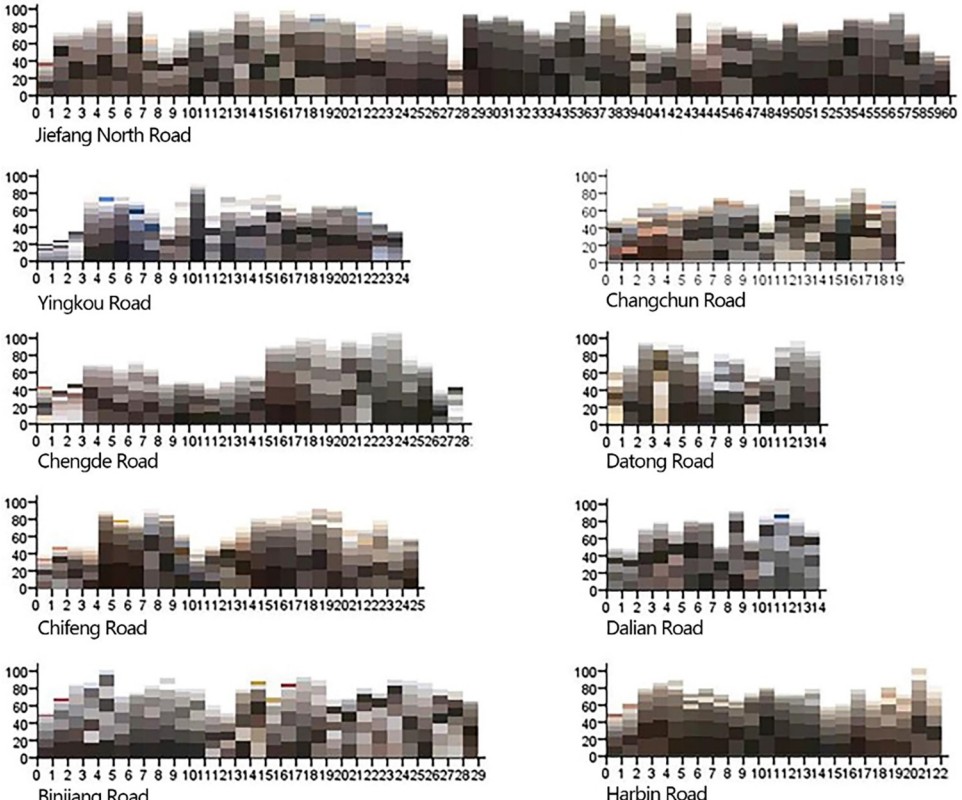

**Fig 14. Architectural color of each viewpoint of each road in Jiefang North Road.**

Road, Binjiang Road, Chengde Road and Datong Road have poor visual continuity. Through the coordinate counter-checking of the viewpoints affecting the continuity, we found that the factors affecting the visual continuity in North Jiefang Road area include the former China-Russia Daosheng Bank Building, the former Macquarie Bank Building and other buildings in the area. The large volume of functional buildings represented by "Rujia Hotel". The non-orderly arrangement of "red bricks", "green and grey decorated bricks" and "dark brown decorated bricks", as well as the width of the road, the openness of the intersection, the height of the building and other surrounding environmental factors.

**Analysis of the main hierarchy of street architectural colors.** The overall dominant color of Jiefang North Road Street is low color value warm gray, the secondary color is high color value warm gray, and the embellishment colors are red-brown, blue-gray, light brown, light yellow and other colors. (**Fig 14**) The dominant colors and auxiliary colors are of the same color scheme and different brightness, and the light brown, bright yellow, brick red and other embellishment colors are close to the dominant colors and auxiliary colors, and the same color tendency makes the overall color of Jiefang North Road Street present a strong assimilation phenomenon. Therefore, the layering of Jiefang North Road Street is not reflected in the saturation of colors, but more in the contrast between warm and cold colors. The neutral and warm color tendency has a sense of forward movement visually, which strengthens the simple and strong visual feeling of the whole Street, while the low-saturation cold gray color gives way to become a background of warm colors, which gives the Jiefang North Road Street a sense of peace and quiet. In addition to the walking sequence in Jiefang North Road, the embellishment colors appear alternately, both with the dominant color of the street color coordination, but

also a certain change, enriching the color landscape of Jiefang North Road. Other roads are easy to see the color of construction, street signs and other non-architectural elements, rather than the design of street space embellishment color, resulting in visual clutter, and the lack of embellishment color richness and appear to be more than the coordination of the lack of hierarchy. This is especially true for Binjiang Road, which is one of the busiest commercial streets in Tianjin and is strictly controlled under the control of the harmonization of historical architecture, with careful use of color, ignoring the ability of color to shape the vitality of commercial space.

## Discussion

### A new method of architectural color measurement based on street view and deep learning

This study realizes the color recognition and efficient evaluation of urban space architecture through the fusion of street view big data and deep learning technology, and provides a large-scale, efficient and low-cost practical method for urban planning and renewal from a more efficient perspective. First, we obtain the street view map of the study area at a high level of detail through the API of Baidu map provider, identify and extract the architectural outline of the urban space by deep learning technology, and identify the architectural color of the whole study area by K-means algorithm. In addition, the approach of using new technologies to intelligently assess the architectural color of urban spaces can help identify potential low-performing areas that need further updated guidance. Specifically, the method can guide the direction of color-related planning strategies based on the color evaluation results and the dominant architectural facade colors of a given street district. The continuous updating of Baidu Street View data also provides reliable data support for the continuous monitoring and updating of architectural colors in urban spaces.

### Architectural color measurement methods offer advice to urban planners

Through the literature, the necessity of studying the color of street space from the perspective of human view and the importance of researching the color of architecture under the street view big data are clarified, and the way of measuring the color of architecture under the street view big data environment is explained in a more complete way. The logical framework of color analysis of street space architecture is explored, which lays the foundation for the automated process of color planning and design of street space architecture, and the combination of theory and technology of model construction makes it possible to transform the color design of street space architecture from special research to universal research. This study takes Jiefang North Road Street in Tianjin as the research object, and effectively practices the measurement of architectural color in street space. The results show that this research method has high applicability and feasibility, and can be widely applied to the basic research of architectural color in other street spaces. A color database was established for Jiefang North Road, and the overall coordination, sequence continuity and sequence hierarchy were analyzed to provide reference and suggestions for the improvement of Jiefang North Road Street.

### Limitations and future work

The dominant color and color value of the street view image data can be affected by weather and times, which is difficult to eliminate by simple color calibration. In the future, we can consider introducing more intelligent methods to improve the color accuracy. The rapid development of computing and information technology has been driving the progress of technology

all the time. In this study, we have combined the deep learning technology and street view data to give an operational tool for measuring the color of urban space and architecture, but at present, due to the limitation of research equipment and hardware, we are unable to expand the scope of the study and automate the whole process. Future research should compare multiple algorithms, find the most efficient algorithm, and make full use of computer performance to expand research ideas and create more efficient research methods. Computer technology-supported color research is a rational and rigorous tool, but architectural color design itself is complex, subject to the constraints of materials, environment, function, regional culture and other factors. This study is based on the analysis of the three elements of color, and the results are indicative, but not comprehensive. For color analysis and evaluation, it still needs the subjective thinking of designers to balance the influencing factors, and designers are still the main character of architectural color research. Whether the future can realize the intelligence of color evaluation is an unknown but worthy of expectation.

## Conclusion

After decades of development, urban color planning has accumulated a lot of experience, but there are also problems that color statistics and research are difficult, and it is difficult to practice on a large coverage, with high efficiency and low cost, especially the research and exploration of architectural color in urban street space. In order to solve this problem, this paper proposes a new method of street space building color measurement based on street view big data and deep learning technology in a pioneering way. The method forms a complete analysis process from the selection of objects, the way of color extraction to the color expression. We first realized the extraction of architectural elements by semantic segmentation of a large number of street images. Then, the street space architectural color data with practical significance were obtained by K-means technique. In order to verify the effectiveness of the proposed method, we also evaluated the research results by using questionnaire analysis. The results show that the proposed method of street space architectural color measurement has high accuracy and feasibility in practical application. We have used the North Jiefang Road in Tianjin as a research site to systematically analyze and evaluate the architectural colors of the street. The new methodology in this paper can assist urban planners and researchers to explore the architectural color of urban spaces more efficiently. The new method in this paper can assist urban planners and research scholars to explore the architectural color of urban space more efficiently. This provides a useful reference for the future architectural color planning and design of urban street space. This will help to realize the harmony and unity of urban space and improve the quality of life of urban residents and the overall image of the city. We will continue to track the development trend of streetscape big data and deep learning technology in order to optimize and improve the street space architectural color measurement method in future research. Through cross research with other fields, we expect to provide more dimensional support for urban space architectural color planning and management, so as to achieve the coordination and sustainability of urban development.

## Author Contributions

**Conceptualization:** Tianlin Zhang.

**Data curation:** Ying Yu.

**Formal analysis:** Ying Yu, Lei Liu, Ming Li, Yingning Shen, Jiazhen Zhang, Xiaomeng Ji, Min Hou.

**Funding acquisition:** Shibao Yu, Hongxu Peng.

**Investigation:** Lei Wang, Yingning Shen, Hongxu Peng.

**Methodology:** Lei Wang, Yingning Shen, Hongxu Peng, Xinpeng Zhang.

**Project administration:** Lei Wang, Hongxu Peng.

**Resources:** Tianlin Zhang, Shibao Yu, Fangzhou Wang, Xinpeng Zhang.

**Software:** Lei Wang, Fengliang Tang, Mingshuai Li, Fangzhou Wang.

**Supervision:** Ying Yu, Tianlin Zhang, Fengliang Tang, Mingshuai Li, Jiazhen Zhang, Xinpeng Zhang.

**Validation:** Ying Yu, Shibao Yu, Fangzhou Wang.

**Visualization:** Fengliang Tang, Mingshuai Li, Fangzhou Wang.

**Writing – original draft:** Xin Han, Lei Wang, Fengliang Tang.

**Writing – review & editing:** Tianlin Zhang, Mingshuai Li, Xiaomeng Ji.

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
