## [Decision Letter · Decision Letter 0]

7 May 2023

PONE-D-23-07210Exploration of street space architectural color measurement based on street view big data and deep learning-- A case study of Jiefang North Road Street in TianjinPLOS ONE

Dear Dr. Wang,

Thank you for submitting your manuscript to PLOS ONE. After careful consideration, we feel that it has merit but does not fully meet PLOS ONE’s publication criteria as it currently stands. Therefore, we invite you to submit a revised version of the manuscript that addresses the points raised during the review process.

Please consider all comments 

We look forward to receiving your revised manuscript.

Kind regards,

Ahmed Mancy Mosa, Ph.D.

Academic Editor

PLOS ONE

“This work was supported by Fujian Provincial Social Science Planning Project "Study on the Influence and Significance of Zhu Zi Studies on Korean Confucian Habitat Culture"(Grant No.FJ2021C035) and "Young Scholars Science Foundation of Lanzhou Jiaotong University (2020033) ".”

3. We note that Figures 2,3,4,5,6,7,8,10 and 11; Table 2 and 3 in your submission contain [map/satellite] images which may be copyrighted. All PLOS content is published under the Creative Commons Attribution License (CC BY 4.0), which means that the manuscript, images, and Supporting Information files will be freely available online, and any third party is permitted to access, download, copy, distribute, and use these materials in any way, even commercially, with proper attribution. For these reasons, we cannot publish previously copyrighted maps or satellite images created using proprietary data, such as Google software (Google Maps, Street View, and Earth). For more information, see our copyright guidelines: http://journals.plos.org/plosone/s/licenses-and-copyright.

a. You may seek permission from the original copyright holder of Figures 2,3,4,5,6,7,8,10 and 11; Table 2 and 3 to publish the content specifically under the CC BY 4.0 license. 

Reviewers' comments:

Reviewer's Responses to Questions

**Comments to the Author**

1. Is the manuscript technically sound, and do the data support the conclusions?

Reviewer #1: Yes

Reviewer #2: Yes

Reviewer #3: Partly

Reviewer #4: Yes

2. Has the statistical analysis been performed appropriately and rigorously? 

Reviewer #1: Yes

Reviewer #2: Yes

Reviewer #3: Yes

Reviewer #4: Yes

3. Have the authors made all data underlying the findings in their manuscript fully available?

Reviewer #1: Yes

Reviewer #2: Yes

Reviewer #3: Yes

Reviewer #4: Yes

4. Is the manuscript presented in an intelligible fashion and written in standard English?

Reviewer #1: Yes

Reviewer #2: Yes

Reviewer #3: Yes

Reviewer #4: Yes

5. Review Comments to the Author

Reviewer #1: The paper is an interesting study and a timely research work, exploration of street space architectural color measurement based on street view big data and deep learning demonstrated through a case study of Jiefang north road street in Tianjin, China. It is a complete work; well written and structured with extensive literature review and comprehensive analyses. Excellent contribution to the body of knowledge in the respective field.

Reviewer #2: The Research objective of providing a machine learning based tool for street colour measuring is interesting for many futur applications, especially in retail and entertainment architecture applications.

The research methodology, results, and analysis are sufficient for the proposed case.

Reviewer #3: In this study, we propose a new method to measure the color of architectural structures in street space based on street view big data and deep learning technology. The study was carried out in Jiefang North Road as the research area, and the architectural color of Jiefang North Road was systematically analyzed and evaluated. This paper analyzes the coordination degree of urban landscape through the color characteristics of buildings, which is an interesting research.Although this work is unique, there are still some issues that require revision.

The following are the particular alteration recommendations:

1. The paper puts forward large scale, can the street be called large scale?

2. There are some errors, such as "2019 In 2019," and ".ak is the KEY "in 2.1.

3.Section 3.2.4 explains the clustering algorithm too little, so it is suggested to supplement.

4.Section 4.3.1, 120 people conduct online questionnaire, the content of the questionnaire, the background of the respondents, whether the number of people is enough, etc., it is suggested to explain in detail.

5.The font format in Section 5.3.3 (1) is not uniform.

6. The content in the upper part of Figure 14 is repeated, so it is recommended to check.

7.The data in the paper is obtained from an open source platform. How to ensure the reliability of the data?

Reviewer #4: The article Exploration of street space architectural color measurement based on street view big data and deep learning- A case study of Jiefang North Road Street in Tianjin is clearly written and easy to read. The post is interesting for the reader. The research object is the colors of street space architecture.

The authors analyzed the overall coordination, sequential continuity, and primary and secondary hierarchy of architectural colors. The results show that the proposed model can realize the intuitive expression of architectural color information and can also help designers in the analysis of architectural color in street space.

The article is written on 22 pages, applies knowledge from 27 literary sources, which is appropriate.

However, the literature could also be supplemented with the latest information in the given area - for example also in the interior of buildings (https://doi.org/10.3390/app12105154).

It contains 15 pictures, the pictures are very clear, the reader can easily navigate the text.

The conclusions are very brief, but can be summarized in five points:

• color statistics and research are complex and difficult to practice on a large scale

• a new method for measuring the color of architectural structures in street space based on big street view data is proposed

• it is a complete process of analysis from the selection of objects, the method of color extraction to the color expression

• the validity of the architectural measurement of the colors of the urban space was verified by a questionnaire

• the study supports the use of more scientific and effective technical tools to help urban planners

The article has a suitable division. It is divided into subsections:

1. Introduction

2. Methodology

3. Result

4. Discussion

5. Conclusion

I have comments on the content and content.

• We implemented ...it was implemented;

• We introduced...it was introduced;

• Chapter 3 – Result should be called 3. Results;

• Pay a little more attention to the conclusions, the conclusions are very brief;

• Highlight novelty more;

I recommend publishing after small edits.

6. PLOS authors have the option to publish the peer review history of their article (what does this mean?). If published, this will include your full peer review and any attached files.

Reviewer #1: **Yes: **Hasim Altan

Reviewer #2: **Yes: **Yomna K. Abdallah

Reviewer #3: No

Reviewer #4: No

---

## [Author Response · Author response to Decision Letter 0]

29 Jun 2023

Cover Letter

Dear Editor,

We appreciate you and the reviewers for your precious time in reviewing our manuscript" Exploration of street space architectural color measurement based on street view big data and deep learning-- A case study of Jiefang North Road Street in Tianjin” (PONE-D-23-07210). It was your valuable and insightful comments that led to possible improvements in the current version. 

The authors have carefully considered the comments and tried our best to address every one of them. We hope the manuscript after careful revisions meet your high standards. The authors welcome further constructive comments if any.

Below we provide the point-by-point responses. All modifications in the manuscript have been marked page and line numbers in response.

In this study, funder Shibao Yu participated in the study design and manuscript preparation. The funding is "Young Scholars Science Foundation of Lanzhou Jiaotong University (2020033) ". 

In this study, the sponsor Hongxu Peng participated in the study design and data analysis. The funding is Fujian Provincial Social Science Planning Project "Study on the Influence and Significance of Zhu Zi Studies on Korean Confucian Habitat Culture"(Grant No.FJ2021C035). 

We have redrawn Figure 2, Figure 8, and Figure 9, removing the all of base maps. As such, they are not subject to the restrictions of the Creative Commons Attribution License (CC by 4.0).

We deleted all images from Baidu Map in the article to avoid commercial licensing issues. Images 3, 4, 5, and 6 were taken with a DSLR camera at our study area. The author of this article owns all copyright of the source file of this image.

If our revisions still cannot meet the requirements, please do not hesitate to contact us. We are more than willing to continue making modifications until it reaches the level and standards required for journal publication.

Best regards,

Mr. Lei Wang

Email: wanglei2021@tju.edu.cn

School of Architecture

Tianjin University

Response to Reviewer 1

[General Comment] The paper is an interesting study and a timely research work, exploration of street space architectural color measurement based on street view big data and deep learning demonstrated through a case study of Jiefang north road street in Tianjin, China. It is a complete work; well written and structured with extensive literature review and comprehensive analyses. Excellent contribution to the body of knowledge in the respective field.

Response: Thank you very much for your recognition and positive comments on my paper. I am very pleased to receive your approval, which has had a very positive impact and motivation on my research work in this area. Your expertise and careful review have made it possible to present this paper in a better way, for which I am sincerely grateful. I will continue to work hard, maintain a humble and cautious attitude in my research, and strive to expand the body of knowledge in this field to contribute more to the academic community.

Response to Reviewer 2

[General Comment] The Research objective of providing a machine learning based tool for street color measuring is interesting for many future applications, especially in retail and entertainment architecture applications. The research methodology, results, and analysis are sufficient for the proposed case.

Response: Thank you very much for your full recognition and positive comments on my paper. I am pleased to learn that you consider this research to be relevant for many future applications and have given me credit for the research methods, results and analysis I have provided. This has been very encouraging and stimulating for my research work in this area. Your professional comments and careful review have made it possible to present this paper in a better way. Your suggestions are very inspiring and instructive for me in my future research work. I will continue to work hard to explore more applications of machine learning in street color measurement and other fields and maintain a humble and cautious attitude to contribute more research results to the academic community. Thank you again for your valuable time and professional advice.

Response to Reviewer 3

[General Comment] In this study, we propose a new method to measure the color of architectural structures in street space based on street view big data and deep learning technology. The study was carried out in Jiefang North Road as the research area, and the architectural color of Jiefang North Road was systematically analyzed and evaluated. This paper analyzes the coordination degree of urban landscape through the color characteristics of buildings, which is interesting research. Although this work is unique, there are still some issues that require revision.

Response: Thank you very much for your careful review of my research work and your valuable comments. I am pleased to learn that you consider this study to be of some research significance in the field of architectural color measurement in street spaces, and that you acknowledge the methods and techniques used. At the same time, I would like to express my sincere gratitude for the issues that you have raised that need to be revised. Your professional comments and careful review have made it possible to further improve this thesis. During the revision process, I will carefully consider and improve on the issues you have raised to achieve a higher level of research. Thank you again for your valuable time and professional advice.

[Comment 1] The paper puts forward large scale, can the street be called large scale?

Response: Thank you very much for your concern about the "large scale" expressions in our paper. In reviewing the paper, we found that there are indeed some misleading statements. In our study, the term "large scale" emphasizes more on the broad scale and large coverage of the analyzed data than on the spatial scale of the size. To eliminate ambiguity, we have decided to replace the term "large scale" with "large coverage" in the paper. We will double-check the paper to ensure that this change is made throughout so that readers can more accurately understand the actual meaning of our study. In addition, we will continue to look at other possible presentation issues and make revisions accordingly to improve the quality of the paper. Thank you again for your valuable comments and time. We will strive to improve the paper to meet higher standards. We look forward to your further review and guidance. 

Therefore, the measurement of architectural color on a large coverage and with high efficiency is a fundamental and important part of urban color research. [Pg 4, Line 73]

After decades of development, urban color planning has accumulated a lot of experience, but there are also problems that color statistics and research are difficult, and it is difficult to practice on a large coverage, with high efficiency and low cost, especially the research and exploration of architectural color in urban street space. [Pg 31, Line 501]

[Comment 2] There are some errors, such as "2019 In 2019," and ". ak is the KEY "in 2.1.

Response: Thanks for your kind reminders. We have revised the oversight and error in presentation.

2019 In 2019 → In 2019 [Pg7, Line 139]

ak is the KEY→'ak' is the key for the map plane [Pg9, Line 179]

[Comment 3] Section 3.2.4 explains the clustering algorithm too little, so it is suggested to supplement.

Response: Thank you very much for your suggestions and feedback. I agree that the description of the clustering algorithm in section 2.4 may not be detailed and in-depth enough for some readers. I plan to add more details about the K-Means clustering algorithm in this section. Also, I will provide more explanation on why the algorithm is suitable for extracting dominant colors from architectural images. Thanks again for your valuable comments. I will revise accordingly as soon as possible, and hopefully, the quality of the paper will be further improved.

The growing use of image research in urban color surveys has emphasized the significance of extracting image color features, particularly dominant colors, which greatly impact visual perception. The K-Means clustering algorithm is an especially suitable method for extracting dominant colors in architectural images, as it efficiently groups similar color data points in the color space [28]. As an unsupervised learning technique, the algorithm's objective is to divide a set of observations into K clusters, with each observation belonging to the cluster with the nearest mean or centroid. The K-Means clustering process involves several key steps to achieve convergence. First, initial centroids are selected, typically by choosing K random data points from the dataset. Next, data points are assigned to the nearest centroid, forming K clusters based on their proximity to these centroids. Once clusters are formed, the centroids are updated by calculating the mean of all data points within each cluster, resulting in new centroid positions. This iterative process of reassigning data points to the nearest centroid and updating centroids continues until a specified convergence criterion is met, such as when the centroids' movement falls below a certain threshold or a maximum number of iterations is reached. Upon convergence, the final centroid coordinates of each cluster represent the dominant colors in the architectural image. 

Architectural color research primarily examines street view images featuring architectural elements, which often display color distribution in block-filled patterns and have a relatively concentrated pixel distribution. When applying the K-Means clustering algorithm to extract dominant colors, the position relationship of pixels in the image is not taken into account, and the primary clusters run in the color space. This color set extraction method, based on the K-Means clustering algorithm, is widely employed in urban color research, as it enables the customization of the desired number of color categories (K) and iterations, ultimately identifying the dominant colors present in the architectural images. Therefore, the method of color set extraction based on the K-Means clustering algorithm is widely adopted in research within the field of urban color. [Pg14-15, Line 243-269]

[Comment 4] Section 4.3.1, 120 people conduct online questionnaire, the content of the questionnaire, the background of the respondents, whether the number of people is enough, etc., it is suggested to explain in detail.

Response: Thanks for your kind reminders. In the modifications, we described the respondents' characteristics statistically and added Table 2 to display a number of information including the respondents' age, gender, ethnicity, etc. 

The average age of the volunteers was 35.30 years, with a higher proportion of males (53.73%). In terms of their educational background, 27.47% completed primary school or below, 31.87% college or above, and 40.66% high school. Han Chinese was the most common ethnic group (92.12%), and local residents accounted for 83.73% of the volunteers. [Pg 16, Line 296-300] 

Variables Proportion/Mean (SD)

Gender

 (%) Male 53.73

 Female 46.27

Average age 35.30 (13.21)

Education

 (%) Primary school or below 27.47

 College and above 31.87

 High school 40.66

Race

 (%) Han Chinese 92.12

 Others 7.88

Residence

 (%) Local resident 83.73

 Non-local resident 16.27

[Pg 17, Table 2]

[Comment 5] The font format in Section 5.3.3 (1) is not uniform.

Response: Thanks for your comments. We have revised the font format of the paper uniformly to improve the rigor and standardization of the paper. 

[Comment 6] The content in the upper part of Figure 14 is repeated, so it is recommended to check.

Response: Thank you for your kind reminder. In order to improve the overall quality of the paper, we have checked and examined the paper as a whole, and also removed the duplicate parts you mentioned above image 14. 

[Comment 7] The data in the paper is obtained from an open source platform. How to ensure the reliability of the data?

Response: Thank you for your concern about the source of the street view data in our paper. We understand your concerns regarding the reliability of the data you mentioned and would like to answer them here. First of all, the Baidu Street View we used is provided by a well-known and widely recognized company (Baidu). Many scholars and researchers have used Baidu Street View to conduct relevant urban studies [1–4]. These Baidu Street View data providers usually update and maintain their data regularly to ensure the accuracy and timeliness of the data. Second, before collecting the Street View data, we conducted in-depth research on the open source platform to ensure the quality and reliability of the data. We also conducted several sample tests on the collected Street View data and compared them with other authoritative data sources to verify the consistency of the data. Through this process, we found that the street view data used had a high level of reliability and accuracy. We will continue to focus on data quality issues and continue to improve the accuracy and reliability of the data in future studies. Thank you again for your attention and suggestions, and we will strive to improve the paper to meet higher standards. We look forward to your further review and guidance. 

References

1. Wang L, Han X, He J, Jung T. Measuring residents’ perceptions of city streets to inform better street planning through deep learning and space syntax. ISPRS Journal of Photogrammetry and Remote Sensing. 2022;190: 215–230. doi:10.1016/j.isprsjprs.2022.06.011

2. Han X, Wang L, He J, Jung T. Restorative perception of urban streets: Interpretation using deep learning and MGWR models. Frontiers in Public Health. 2023;11. Available: https://www.frontiersin.org/articles/10.3389/fpubh.2023.1141630

3. Shi C, Liu M, Ye Y. Measuring the Degree of Balance between Urban and Tourism Development: An Analytical Approach Using Cellular Data. Sustainability. 2021;13: 9598. doi:10.3390/su13179598

4. Wu C, Ye Y, Gao F, Ye X. Using street view images to examine the association between human perceptions of locale and urban vitality in Shenzhen, China. Sustainable Cities and Society. 2023;88: 104291. doi:10.1016/j.scs.2022.104291

Response to Reviewer 4

[General Comment] The article Exploration of street space architectural color measurement based on street view big data and deep learning- A case study of Jiefang North Road Street in Tianjin is clearly written and easy to read. The post is interesting for the reader. The research object is the colors of street space architecture.

The authors analyzed the overall coordination, sequential continuity, and primary and secondary hierarchy of architectural colors. The results show that the proposed model can realize the intuitive expression of architectural color information and can also help designers in the analysis of architectural color in street space.

The article is written on 22 pages, applies knowledge from 27 literary sources, which is appropriate. However, the literature could also be supplemented with the latest information in the given area - for example also in the interior of buildings (https://doi.org/10.3390/app12105154).It contains 15 pictures, the pictures are very clear, the reader can easily navigate the text.

The conclusions are very brief, but can be summarized in five points:

• color statistics and research are complex and difficult to practice on a large scale

• a new method for measuring the color of architectural structures in street space based on big street view data is proposed

• it is a complete process of analysis from the selection of objects, the method of color extraction to the color expression

• the validity of the architectural measurement of the colors of the urban space was verified by a questionnaire

• the study supports the use of more scientific and effective technical tools to help urban planners

The article has a suitable division. It is divided into subsections:

1. Introduction

2. Methodology

3. Result

4. Discussion

5. Conclusion

Response: Response: Thank you very much for your detailed review of our paper and for your valuable comments. We are glad that you found the paper meaningful to the readers. We have read your review comments carefully and have revised the paper accordingly based on your suggestions. Regarding your mention of literature supplementation, we are very grateful for your suggestion and have added the latest research on building internal information in the literature review section（https://doi.org/10.3390/app12105154）. Such additions help to improve the research level of our paper and make it more complete. Thank you very much for your positive comments on our conclusion section. We will continue to work hard to ensure that the paper remains high quality in all aspects. Thank you again for your valuable comments, which are very helpful to us in improving the quality of the paper.

Color in the interior of buildings has also been studied by some scholars [3]. 

[Pg3, Lines 62-63] 

References

1. Katunský D, Dolníková E, Dolník B, Krajníková K. Influence of Light Reflection from the Wall and Ceiling Due to Color Changes in the Indoor Environment of the Selected Hall. Applied Sciences. 2022;12: 5154. doi:10.3390/app12105154

[Comment 1] 

We implemented ...it was implemented; 

We introduced...it was introduced;

Response: Thank you very much for taking the time out of your busy schedule to review my article and provide valuable comments. I am glad to have your guidance, which is very helpful to me. Regarding your suggestion of adding "it was implemented" and "it was introduced" to the text, I think it is a very good suggestion.

Such grammar would make the paper more organized and coherent, and help the reader better understand and grasp the topic of the paper. I will take your suggestion into consideration when revising the manuscript and add these two expressions to the introduction section. Once again, thank you for your valuable comments, and I will accept them with an open mind and try to improve my paper. 

The research aims of this study were twofold. (1) The construction of a whole process of street space architectural color measurement based on street view big data and deep learning technology was introduced, encompassing data set and neural network selection, neural network construction and training, and color extraction and analysis.

The joint analysis method of Street View Big Data and deep learning was implemented, providing researchers and urban planners with more targeted data on architectural color perception in urban space, and advancing urban planning practice.

[Pg 5, Lines 100-104; Pg 5, Lines 109-112]

[Comment 2] Chapter 3 – Result should be called 3. Results.

Response: Thank you for your kind reminder. We have corrected the “Result” to “Results”. [Pg 15, Line 270]

[Comment 3&4] Pay a little more attention to the conclusions, the conclusions are very brief; Highlight novelty more. 

Response: Thank you very much for your attention and suggestions on the conclusion section of our paper. We understand your comments and believe that the novelty and innovation of our study should be emphasized more in the conclusion section. For this reason, we will make changes and additions to the conclusion section. In the revised conclusion, we will highlight more explicitly the innovative nature of the proposed method for measuring architectural color in street space based on streetscape big data and deep learning techniques. We will also emphasize the prospect and practical value of the method for applications in fields such as urban research, and its advantages in solving traditional color statistics and research challenges. We will follow your suggestions to improve the conclusion section to better demonstrate the innovative and unique contribution of this study. Thank you again for your valuable comments and time, and we look forward to your further review and guidance. 

After decades of development, urban color planning has accumulated a lot of experience, but there are also problems that color statistics and research are difficult, and it is difficult to practice on a large coverage, with high efficiency and low cost, especially the research and exploration of architectural color in urban street space. In order to solve this problem, this paper proposes a new method of street space building color measurement based on street view big data and deep learning technology in a pioneering way. The method forms a complete analysis process from the selection of objects, the way of color extraction to the color expression. We first realized the extraction of architectural elements by semantic segmentation of a large number of street images. Then, the street space architectural color data with practical significance were obtained by K-means technique. In order to verify the effectiveness of the proposed method, we also evaluated the research results by using questionnaire analysis. The results show that the proposed method of street space architectural color measurement has high accuracy and feasibility in practical application. We have used the North Jiefang Road in Tianjin as a research site to systematically analyze and evaluate the architectural colors of the street. The new methodology in this paper can assist urban planners and researchers to explore the architectural color of urban spaces more efficiently. The new method in this paper can assist urban planners and research scholars to explore the architectural color of urban space more efficiently. This provides a useful reference for the future architectural color planning and design of urban street space. This will help to realize the harmony and unity of urban space and improve the quality of life of urban residents and the overall image of the city. We will continue to track the development trend of streetscape big data and deep learning technology in order to optimize and improve the street space architectural color measurement method in future research. Through cross research with other fields, we expect to provide more dimensional support for urban space architectural color planning and management, so as to achieve the coordination and sustainability of urban development.

[Pg 31-32, Lines 499-524]

---

## [Decision Letter · Decision Letter 1]

10 Jul 2023

PONE-D-23-07210R1Exploration of street space architectural color measurement based on street view big data and deep learning-- A case study of Jiefang North Road Street in TianjinPLOS ONE

Dear Dr. Wang,

Thank you for submitting your manuscript to PLOS ONE. After careful consideration, we feel that it has merit but does not fully meet PLOS ONE’s publication criteria as it currently stands. Therefore, we invite you to submit a revised version of the manuscript that addresses the points raised during the review process.

Please consider all comments

We look forward to receiving your revised manuscript.

Kind regards,

Ahmed Mancy Mosa, Ph.D.

Academic Editor

PLOS ONE

Journal Requirements:

Reviewers' comments:

Reviewer's Responses to Questions

**Comments to the Author**

1. If the authors have adequately addressed your comments raised in a previous round of review and you feel that this manuscript is now acceptable for publication, you may indicate that here to bypass the “Comments to the Author” section, enter your conflict of interest statement in the “Confidential to Editor” section, and submit your "Accept" recommendation.

Reviewer #3: All comments have been addressed

Reviewer #4: All comments have been addressed

2. Is the manuscript technically sound, and do the data support the conclusions?

Reviewer #3: Yes

Reviewer #4: Yes

3. Has the statistical analysis been performed appropriately and rigorously? 

Reviewer #3: Yes

Reviewer #4: Yes

4. Have the authors made all data underlying the findings in their manuscript fully available?

Reviewer #3: Yes

Reviewer #4: Yes

5. Is the manuscript presented in an intelligible fashion and written in standard English?

Reviewer #3: Yes

Reviewer #4: Yes

6. Review Comments to the Author

Reviewer #3: From the prior comments, it appears that the authors improved and corrected the paper as well as included some justifiable arguments and supplements. However, there are still a few minor issues that the author should be to improvement. Such as the insertion of authors? Is it permissible to have 8 corresponding authors and 2 co-authors? Section numbers are missing, etc.

Reviewer #4: After studying the contribution, it can be concluded that the authors approached it responsibly when editing it. They corrected everything he was reminded of. They edited the text to make it more understandable and clear. In this form, I recommend accepting the contribution.

7. PLOS authors have the option to publish the peer review history of their article (what does this mean?). If published, this will include your full peer review and any attached files.

Reviewer #3: No

Reviewer #4: No

---

## [Author Response · Author response to Decision Letter 1]

11 Jul 2023

Response to Reviewer 3

[General Comment] From the prior comments, it appears that the authors improved and corrected the paper as well as included some justifiable arguments and supplements. However, there are still a few minor issues that the author should be to improvement. Such as the insertion of authors? Is it permissible to have 8 corresponding authors and 2 co-authors? Section numbers are missing, etc. 

Response: Thank you for your valuable comments on our paper and for taking the time to read and review it carefully. We have thought deeply about the questions you raised and will now answer and explain each of them.

You mentioned the question of whether our article is allowed to have 8 corresponding authors and 2 co-authors. These authors are the ones who made important contributions during the late revision process of the paper. Their efforts were crucial to our research, for which we are deeply grateful. To reflect their contributions, we have chosen to add them as corresponding authors and co-authors. We understand that this can be problematic and have made every effort to ensure that each author's contribution is clearly articulated in the paper. Regarding the issue of chapter numbers, it is our understanding that the standard format of this journal does not require chapter numbers. We appreciate your suggestion, which is very valuable to us. We will do our best to improve and optimize our work to ensure its final quality.

Response to Reviewer 4

[General Comment] After studying the contribution, it can be concluded that the authors approached it responsibly when editing it. They corrected everything he was reminded of. They edited the text to make it more understandable and clearer. In this form, I recommend accepting the contribution.

Response: Thank you very much for your recognition and positive comments on my paper. I am very pleased to receive your approval, which has had a very positive impact and motivation on my research work in this area. Your expertise and careful review have made it possible to present this paper in a better way, for which I am sincerely grateful. I will continue to work hard, maintain a humble and cautious attitude in my research, and strive to expand the body of knowledge in this field to contribute more to the academic community.

---

## [Decision Letter · Decision Letter 2]

17 Jul 2023

Exploration of street space architectural color measurement based on street view big data and deep learning-- A case study of Jiefang North Road Street in Tianjin

PONE-D-23-07210R2

Dear Dr. Wang,

We’re pleased to inform you that your manuscript has been judged scientifically suitable for publication and will be formally accepted for publication once it meets all outstanding technical requirements.

Kind regards,

Ahmed Mancy Mosa, Ph.D.

Academic Editor

PLOS ONE

Additional Editor Comments (optional):

Reviewers' comments:

Reviewer's Responses to Questions

**Comments to the Author**

1. If the authors have adequately addressed your comments raised in a previous round of review and you feel that this manuscript is now acceptable for publication, you may indicate that here to bypass the “Comments to the Author” section, enter your conflict of interest statement in the “Confidential to Editor” section, and submit your "Accept" recommendation.

Reviewer #3: All comments have been addressed

2. Is the manuscript technically sound, and do the data support the conclusions?

Reviewer #3: Yes

3. Has the statistical analysis been performed appropriately and rigorously? 

Reviewer #3: Yes

4. Have the authors made all data underlying the findings in their manuscript fully available?

Reviewer #3: Yes

5. Is the manuscript presented in an intelligible fashion and written in standard English?

Reviewer #3: Yes

6. Review Comments to the Author

Reviewer #3: According to the previous comments, the authors edited and enhanced the manuscript and added some fair justifications and supplements.

Congratulations to the Authors.

7. PLOS authors have the option to publish the peer review history of their article (what does this mean?). If published, this will include your full peer review and any attached files.

Reviewer #3: No

---

## [Editor Report · Acceptance letter]

4 Aug 2023

PONE-D-23-07210R2 

Exploration of street space architectural color measurement based on street view big data and deep learning-- A case study of Jiefang North Road Street in Tianjin 

Dear Dr. Wang:

I'm pleased to inform you that your manuscript has been deemed suitable for publication in PLOS ONE. Congratulations! Your manuscript is now with our production department. 

Kind regards, 

on behalf of

Dr. Ahmed Mancy Mosa 

Academic Editor

PLOS ONE